# Natural Methoxyphenol Compounds: Antimicrobial Activity against Foodborne Pathogens and Food Spoilage Bacteria, and Role in Antioxidant Processes

**DOI:** 10.3390/foods10081807

**Published:** 2021-08-05

**Authors:** Elena Orlo, Chiara Russo, Roberta Nugnes, Margherita Lavorgna, Marina Isidori

**Affiliations:** Department of Environmental, Biological and Pharmaceutical Sciences and Technologies, University of Campania “Luigi Vanvitelli”, Via Vivaldi 43, 81100 Caserta, Italy; elena.orlo@unicampania.it (E.O.); roberta.nugnes@unicampania.it (R.N.); margherita.lavorgna@unicampania.it (M.L.); marina.isidori@unicampania.it (M.I.)

**Keywords:** natural compounds, antimicrobial activity, antioxidant activity

## Abstract

The antibacterial and antioxidant activities of three methoxyphenol phytometabolites, eugenol, capsaicin, and vanillin, were determined. The in vitro antimicrobial potential was tested on three common foodborne pathogens (*Escherichia coli*, *Pseudomonas aeruginosa*, *Staphylococcus aureus*) and three food spoilage bacteria (*Shewanella putrefaciens*, *Brochothrix thermosphacta*, and *Lactobacillus plantarum*). The antioxidant assays were carried out for studying the free radical scavenging capacity and the anti-lipoperoxidant activity. The results showed that eugenol and capsaicin were the most active against both pathogens and spoilage bacteria. *S. aureus* was one of the most affected strains (median concentration of growth inhibition: IC50 eugenol = 0.75 mM; IC50 capsaicin = 0.68 mM; IC50 vanillin = 1.38 mM). All phytochemicals slightly inhibited the growth of *L. plantarum*. Eugenol was the most active molecule in the antioxidant assays. Only in the oxygen radical absorbing capacity (ORAC) test did vanillin show an antioxidant activity comparable to eugenol (eugenol ORAC value = 2.12 ± 0.08; vanillin ORAC value = 1.81 ± 0.19). This study, comparing the antimicrobial and antioxidant activities of three guaiacol derivatives, enhances their use in future applications as food additives for contrasting both common pathogens and spoilage bacteria and for improving the shelf life of preserved food.

## 1. Introduction

Nutraceuticals include a wide range of bioactive compounds, such as antioxidants, fatty acids, phytochemicals, and amino acids, with recognized anti-inflammatory, anti-ageing, and anti-cancer properties [1,2]. In detail, phytochemicals, including flavonoids, phenylpropanoids, polyphenolics, and terpenoids, represent the main class of nutraceuticals with a wide spectrum of beneficial activities for human health.

Nowadays, a growing interest is also turning towards the use of phytochemicals in the prevention of food diseases as well as microbial and oxidative food deterioration, avoiding the reduction of food nutritional value and shelf life [3] and possible losses of environmental and economic resources. This interest arises from the need to replace conventional synthetic compounds such as butylated hydroxytoluene (BHT) and butylated hydroxyanisole (BHA), considered by the European Commission as feed additives since 1987 [4] but causing harmful effects on human health [5]. The use of natural substances, in addition to the typical role in food preservation of the synthetic compounds, would provide the advantages of nutraceuticals.

The antimicrobial and antioxidant activities of three methoxyphenol phytometabolites, eugenol, capsaicin, and vanillin (Table 1), which are normally present in edible plants, have long been recognized, and they have been studied as alternative strategies to prevent food deterioration due to their safety and nontoxic status.

Eugenol is a phenylpropanoid derived from guaiacol with an allyl chain substituted para to the hydroxy group, well known for the aromatic oil extracted from cloves, with antiviral, analgesic, and anti-inflammatory properties [6]. Capsaicin is the major alkaloid of chili peppers, used in folk medicine for its analgesic, antioxidant, anti-inflammatory, and anticancer activities [7]. Vanillin is a phenolic aldehyde, known as the primary component of the extract of the vanilla bean. It is generally recognized as safe (GRAS), has anticancer properties, and can work as a relaxing and calming compound that acts on the nervous system [8,9,10,11,12].

The three guaiacol derivatives have been extensively in vitro tested against a broad spectrum of foodborne pathogens and food spoilage bacteria, and studied for their antioxidant properties [13]. Nevertheless, considering antimicrobial and antioxidant activities as a hurdle concept, studies on the complementary food preservative approach are still few [14,15]. Food is subjected to a long chain of processing, where each stage is a potential source of contamination. Secondary contamination (especially handling and surface contamination) and tertiary contamination (especially retail) could lead to the presence of common bacteria, such as coliforms, cocci, etc., and the development of spoilage bacteria causing undesirable effects on the organoleptic product quality with consequent food loss and foodborne diseases [16].

In light of the above, in the present study, we evaluated the in vitro antimicrobial potential of the selected phytomolecules using three representative common foodborne pathogens: *Escherichia coli*, *Pseudomonas aeruginosa*, and *Staphylococcus aureus*, as well as three representative food spoilage bacteria: *Shewanella putrefaciens*, *Brochothrix thermosphacta*, and *Lactobacillus plantarum*, frequently involved, respectively, in food contamination and in food spoilage. In addition to the critical role that microbial deterioration plays in the loss of food quality, lipid peroxidation and oxidative processes also lead to alterations in the flavor, texture, and color of food, thus influencing the acceptance of consumers [3].

Hence, in this research, a battery of antioxidant assays was carried out in order to study the free radical scavenging capacity as well as the anti-lipoperoxidant activity of eugenol, capsaicin, and vanillin in order to obtain results useful in the development of strategies to maintain food nutritional quality and enhance food shelf life.

## 2. Materials and Methods

### 2.1. Reagents

Ethanol (EtOH, CAS: 64-17-5), methanol (CAS: 67-56-1), 2,2-diphenyl-1-picrylhydrazyl (DPPH, CAS: 1898-66-4), 2′-azino-bis (3-ethylbenzothiazoline-6-sulfonic acid) diammonium salt (ABTS, CAS: 30931-67-0), potassium persulfate (CAS: 7727-21-1), 6-hydroxy-2,5,7,8-tetramethylchroman-2-carboxylic acid (Trolox, CAS: 53188-07-1), (±)-α-tocopherol (CAS: 10191-41-0), Tween 80 (CAS: 9005-66-7), 2-thiobarbituric acid (TBA, CAS: 504-17-6), tannic acid (TA, CAS:1401-55-4) trichloroacetic acid (TCA, CAS: 76-03-9), tris hydrochloride (CAS: 1185-53-1), rapeseed oil (CAS: 8002-13-9), fluorescein (CAS: 518-47-8), 2,2′-azobis(2-methylpropionamidine) dihydrochloride (AAPH, CAS: 2997-92-4), and 7-hydroxy-3H-phenoxazin-3-one-10-oxide sodium salt (resazurin, CAS: 62758-13-8) were supplied by Sigma-Aldrich (Milano, Italy).

### 2.2. Phytochemicals

Eugenol (purity ≥ 99%, CAS: 97-53-0), capsaicin (purity ≥ 95%, CAS: 404-86-4), and vanillin (purity ≥ 97%, CAS: 121-33-5) (Sigma-Aldrich, Milano, Italy) stock solutions were respectively prepared at 500 mM (20% EtOH), 125 mM (100% EtOH), and 500 mM (20% EtOH), under sterile conditions.

### 2.3. Bacterial Strains and Growth Conditions

Three Gram-negative bacterial strains, *Escherichia coli* (ATCC 13762), *Pseudomonas aeruginosa* (ATCC 9027), and *Shewanella putrefaciens* (ATCC 8071), and three Gram-positive bacterial strains, *Staphylococcus aureus* (ATCC 6538), *Brochothrix thermosphacta* (ATCC 11509), and *Lactobacillus plantarum* (WCFS 1), were used. The foodborne pathogens *E. coli*, *P. aeruginosa*, *S. aureus* as well as the food spoilage *S. putrefaciens* were cultivated respectively at 37 °C and 26 °C, in Tryptic Soy Broth (TSB) and Tryptic Soy Agar (TSA) (Oxoid, Milano, Italy). The food spoilage bacteria *B. thermosphacta* and *L. plantarum* were cultivated respectively at 26 °C in Brain-Heart Infusion and Brain-Heart Agar (BHI and BHA, Conda-Pronadisa, Madrid, Spain), and at 30 °C in de Man, Rogosa and Sharpe (MRS) broth and MRS agar (MRSA) (Sigma Aldrich, St. Louis, Switzerland).

For the experiments, cultures were sterilely prepared, picking one colony of each bacterial strain from an agar plate with a loop and then inoculating it overnight in 10 mL of the corresponding medium at the specific growth temperature (reported above).

### 2.4. Antibacterial Activity

#### 2.4.1. Microbial Susceptibility Assay

The broth-based turbidimetric assay was performed in line with Kolarević et al. [17] and Lavorgna et al. [18]. The assay was performed using 96-well flat bottom microtiter plates (Sarstedt, Milano, Italy) prepared in quadruplicate as follows: 100 µL of medium, 100 µL of phytochemicals at different concentrations, and 100 µL of the bacterial culture corresponding to 1 × 10^4^ colony-forming units (CFU) in each well.

Phytochemical concentrations were prepared in 0.9% sterile physiological solution and were chosen after specific range finding tests for each bacterial strain. In particular, eugenol dilutions were prepared from 40 to 0.312 mM (dilution factor, DF of 2) for *L. plantarum* exposure, from 2.5 to 0.039 mM (DF = 2) for *S. putrefaciens*, and from 10 to 0.039 mM (DF = 2) for the exposure of all other bacterial strains. Capsaicin dilutions were prepared from 10 to 0.039 mM for all bacterial treatments. Finally, vanillin dilutions were prepared from 40 to 0.312 mM, from 20 to 0.039 mM (DF = 2), and from 5 to 0.039 mM (DF = 2), respectively, for *L. plantarum*, *B. thermosphacta*, and *S. putrefaciens* treatments, as well as from 10 to 0.039 mM for the exposure of all other bacterial strains.

In addition, 100 μL of 0.9% NaCl as well as 100 μL of streptomycin/ampicillin (0.03 mM) were respectively used as negative control and positive control. Moreover, to guarantee no antimicrobial activity exerted by ethanol, solvent controls were prepared (from 8% to 0.1%). For each plate, a blank (only medium) was prepared in triplicate. Plates were incubated for 24 h at the corresponding growth bacteria temperature. The optical density of each well at 600 nm was recorded using a microplate reader (Synergy H1, Biotek, Winooski, VT, USA). The results were calculated as inhibition of cell bacteria growth (IC) percentage according to the following formula:IC % = [1 − (OD sample/OD negative control)] × 100(1)

#### 2.4.2. MIC Assay

The assay, performed according to Nikolić and co-authors [19], was carried out in 96-well microtiter plates by making serial dilutions (DF = 2) of the phytochemicals in the corresponding media. Specifically, eugenol dilutions were from 100 to 0.78 mM for *L. plantarum* exposure, from 25 to 0.19 mM for *E. coli*, *P. aeruginosa*, *and B. thermosphacta*, from 12.5 to 0.098 mM for *S. aureus*, and from 6.25 to 0.049 mM for *S. putrefaciens*. Capsaicin dilutions were prepared from 12.5 to 0.098 mM for all bacterial treatments. Finally, vanillin dilutions were prepared from 100 to 0.78 mM for *L. plantarum*, from 50 to 0.39 mM for *P. aeruginosa* and *B. thermosphacta*, from 25 to 0.19 mM *E. coli* and *S. aureus*, and from 12.5 to 0.098 mM for *S. putrefaciens*.

Ampicillin or streptomycin were used as positive controls at 0.03 mM, while 0.9% NaCl was used as negative control, and 200 µL of only medium was used to prepare the blank. Ethanol (0.25–10%) was used for solvent control. In each well, 20 µL of bacteria was inoculated to have 1 × 10^4^ CFU with the final volume equal to 200 µL. Plates were incubated for 24 h at the corresponding growth bacteria temperature. For each bacterial strain, each test was prepared in quadruplicate.

In order to have MIC values, 22 µL of aqueous solution of resazurin (0.675 mg/mL), an indicator of cell growth, was added to each well, and the plates were incubated for 3 h. The lowest concentration at which no visible color change occurred was taken as the minimal inhibitory concentration (MIC) value. Whether the effect was bacteriostatic or bactericidal was established by plating samples from wells without visible growth onto respective agar medium. Minimal bactericidal concentration (MBC) was the lowest concentration that showed no visible growth onto agar plate after incubation for 24–48 h at the corresponding growth bacteria temperature. For each tested compound and bacterial strain, the MIC and the MBC values were the average of the values coming from three independent experiments.

### 2.5. Antioxidant Activity

#### 2.5.1. DPPH Assay

DPPH• scavenging capability was recorded according to Brand-Williams and co-authors [20], using 96-well microtiter plates. Phytochemical concentrations, chosen after specific range finding tests, were prepared in sterile deionized water. Next, 15 μL of each compound dilution (0.019–0.3 mM of eugenol and capsaicin; 0.3–9.6 mM of vanillin, DF = 2) was mixed with 235 μL of DPPH• methanol solution (101.43 μM) at room temperature.

Trolox^®^ was used as standard antioxidant. After 30 min, the absorbance was measured at 515 nm in reference to a negative control (15 μL of distilled water in 235 μL of DPPH• solution). DPPH• scavenging percentage (RS%) was calculated as follows:RS (%) = [(OD negative control − OD sample)/(OD negative control)] × 100(2)

Each test was prepared in quadruplicate. The results were expressed as IC50 (the concentration of phytochemicals to scavenge the DPPH• of 50%) values and as TEAC (Trolox^®^ Equivalent Antioxidant Capacity) values calculated according to the following formula:TEAC = IC50 Trolox^®^/IC50 sample(3)

#### 2.5.2. ABTS Assay

The antiradical activity of phytochemicals was also assessed using ABTS method performed according to Lavorgna et al. [21] using 96-well microtiter plates. Potassium persulphate (140 mM) was added to the aqueous solution of ABTS (7 mM) and then incubated for 12–16 h in the dark at room temperature to generate the radical solution ABTS•+. The mixture was diluted with PBS to obtain an absorbance of 0.7 ± 0.02 OD at 734 nm.

Compound concentrations, prepared in distilled water, were chosen after range finding tests and were equal to 0.004–0.03 mM for eugenol and capsaicin, as well as equal to 0.94–15.0 mM for vanillin (DF = 2). Hence, 22.7 μL of compound concentrations or Trolox^®^ were mixed with 227 μL of ABTS•+ solution (101.43 μM) at room temperature. After 6 min, the absorbance was measured at 734 nm in reference to a negative control (22.7 μL of distilled water in 227 μL of ABTS•+ solution).

RS% and TEAC values were calculated using Equations (2) and (3).

#### 2.5.3. TBARS Assay

A thiobarbituric acid reactive substances (TBARS) assay was carried out according to the method of Sroka and Cisowski [22]. This assay allows the end products of polyunsaturated fatty acid oxygenation to be determined when they react with thiobarbituric acid and form red adducts. TBA reagent was prepared by mixing 30 mL of Reagent A (375 mg of TBA and 30 mg of TA dissolved in 30 mL of hot water) and 70 mL of Reagent B (15 g of TCA dissolved in 70 mL of 0.3 M HCl aqueous solution). Next, 5.2 µL of rapeseed oil was emulsified with 15.6 mg of Tween-40 dissolved in 2 mL of 0.2 M Tris-HCl buffer (pH 7.4). Then, emulsion was irradiated at 254 nm with UV light at room temperature for 1 h. Phytochemical concentrations (0.006–0.750 mM of eugenol; 0.006–3 mM of capsaicin; 0.012–3 mM of vanillin, DF = 2) were prepared in distilled water and were chosen after range finding tests. Thus, 100 µL of each test compound concentration, tested in quadruplicate, was added to 1 mL of reaction mixture and then irradiated with UV light for 30 min.

Subsequently, 2 mL of TBA was added to the irradiated samples and tubes were placed in boiling water for 15 min and centrifuged at 1500× *g* for 3 min. The supernatant was measured at 532 nm. Results were expressed as the ability percentage of the phytochemicals to inhibit lipid peroxidation (ILP%) using a negative control, in line with the following equation:ILP (%) = [(OD negative control − OD sample)/(OD negative control)] × 100(4)

The registered activity by each test compound was compared to those of α-tocopherol, used as standard in this assay.

#### 2.5.4. ORAC Assay

An oxygen radical absorbing capacity (ORAC) assay was performed as described by Gillespie and co-authors [23]. First, 25 µL of eugenol, capsaicin, and vanillin (0.09–6.0 µM, DF = 2, chosen after range finding tests and prepared in phosphate buffer solution, PBS, 75 mM, pH 7) and 150 µL of fluorescein (FL, 80 nM prepared in PBS) were co-incubated for 10 min at 37 °C. Then, 25 µL of AAPH solution (150 mM) was added.

In parallel, a blank (FL + AAPH) and the standard antioxidant Trolox^®^ (1–8 µM) were properly prepared in PBS. The fluorescence decay (λex = 485 nm, λem = 525 nm) was recorded every 5 min for 55 min using a fluorescence reader. The area under the fluorescence decay curve (AUC) was calculated as follows:AUC = (0.5 + F5/F0 + F10/F0 + F15/F0 + …… + Fn/F0) × 5(5)
where F0 = fluorescence reading at time 0; Fn = fluorescence reading at time n.

From the AUC of the blank and AUC of phytochemicals, net AUC was calculated using the following equation:AUCnet = AUCsample − AUCblank(6)

The antioxidant activity (ORAC value) in terms of µM Trolox with equivalent activity to 1 µM phytochemical (or Trolox equivalents: µM TE) was calculated as suggested by Sharpe et al. [24] by using the Trolox^®^ calibration curve as well as phytochemical curves (AUCnet/µM) to obtain the slope (m) values: (slope sample/slope Trolox). Furthermore, the concentration (µM) of the phytochemical to scavenge the oxygen radical to guarantee 50% (IC50) of the fluorescence maintenance after 55 min was calculated.

### 2.6. Statistical Analysis

For each assay, three independent experiments were interpolated and IC50 values were obtained by Prism 5 analysis (GraphPad Inc., San Diego, CA, USA), using non-linear regression (log agonist vs. normalized response-variable slope) with 95% confidence range. Lowest observed adverse effect concentrations (LOAECs) were calculated by one-way ANOVA Dunnett’s comparison test, and differences from negative controls or from standards were considered significant as follows: * *p* < 0.05, ** *p* < 0.001, and *** *p* < 0.0001.

Significant differences (*p* < 0.05) among samples were calculated by one-way ANOVA, Tukey’s multiple comparison test, and expressed using different letters. Both for antimicrobial and antioxidant assays, the statistically significant increase of activities was analyzed by two-way ANOVA Bonferroni post-tests (*p* < 0.05) considering the variations among molecules, strains, or assays.

## 3. Results

### 3.1. Antibacterial Activity

Eugenol, capsaicin, and vanillin were tested against three foodborne pathogens, *E. coli*, *P. aeruginosa*, and *S. aureus*, and three food spoilage bacteria, *S. putrefaciens*, *B. thermosphacta*, and *L. plantarum*. Concentrations inhibiting 50% of bacterial growth (IC50s) are reported in Table 2.

In general, eugenol and capsaicin were the most active phytochemicals against bacterial strains with a broader spectrum of action. In detail, these compounds were statistically (one-way ANOVA Bonferroni post-test analysis) more effective on all Gram-negative strains (IC50 values ranging from 1.11 to 2.70 mM and from 1.21 to 4.79 mM, respectively, for eugenol and capsaicin) as well as on the Gram-positive strain *S. aureus* (IC50 eugenol = 0.75 mM; IC50 capsaicin = 0.68 mM). Differently, vanillin was more active on *S. putrefaciens* (Gram-negative, IC50 = 2.60 mM) and *S. aureus* (Gram-positive, IC50 = 1.38 mM).

On the other hand, all phytochemicals inhibited the growth of *L. plantarum* at high concentrations in the order of dozens of mmol/L, highlighting the highest resistance of this microorganism to the selected natural substances. When testing capsaicin, only a 25.96% of inhibition cell growth was obtained at the highest concentration tested (10 mM) without reaching IC50 value. Because of the low solubility of capsaicin, it was not possible to test higher concentrations to avoid solvent (ethanol) effective percentages, as revealed by testing solvent controls.

To better understand how the three phytochemicals influenced the exponential microbial growth, concentration-effect curves were drawn (Figure 1).

Basically, the curves showed evident concentration–response effects for all tested substances, with different trends in line with the bacterial strain. In particular, *S. aureus* curves were always characterized by a moderate increase in concentration/effect relationship, while those of *S. putrefaciens* showed a rapid increase with a higher effect at lower concentrations when treated with eugenol and vanillin. A similar sharp trend, but at high concentrations, was observed when testing compounds on *L. plantarum* and *B. thermosphacta*. Among phytochemicals, the highest microbial growth inhibition percentages (~80–90%) were reached testing eugenol; specifically, a high effect (91.72%) was obtained on *S. putrefaciens* at the lowest concentration of 2.5 mM. In addition, when eugenol was tested on *B. thermosphacta*, a higher antimicrobial activity (88.26% at 10 mM) than capsaicin (74.81% at 10 mM) and vanillin (83.35% at 20 mM) was registered.

Apart from the highest effect concentrations (just described), the curves together with the statistical analysis (one-way ANOVA, Dunnett’s multiple comparison test) allowed us to study the lowest observed adverse effect concentrations (LOAECs) (Figure 2).

The lowest LOAEC (0.078 mM) was obtained exposing *S. aureus* to all three phytochemicals, and *S. putrefaciens* to eugenol. Capsaicin (on *E. coli* and *L. plantarum*) and vanillin (on *B. thermosphacta*) showed LOAEC values equal to 0.625 mM, lower than eugenol.

Bacteriostatic and bactericidal activities were also studied, and results, expressed as minimal inhibitory concentration (MIC) and minimal bactericidal concentration (MBC), are reported in Table 3.

The lowest MICs equal to 3.125 and 6.25 mM, and MBCs 6.25 and 12.5 mM were obtained testing eugenol on *S. putrefaciens* and *S. aureus*, respectively. On the contrary, *L. plantarum* was the most resistant strain to all phytomolecules (especially for vanillin). Capsaicin MIC values (12.5 mM) were found only for *S. putrefaciens* and *S. aureus*, while it was not possible to establish a specific value for all other bacteria because of the solvent activity interference as explained above.

### 3.2. Antioxidant Activity

Results of DPPH, ABTS, and TBARS assays were expressed as the concentration of phytochemicals able to scavenge the DPPH or ABTS radicals, or to inhibit the lipid peroxidation by 50% (IC50), and are reported in Table 4.

Eugenol and capsaicin showed good antiradical activity without statistically significant differences between them (one-way ANOVA, Tukey’s multiple comparison test) in scavenging both DPPH and ABTS radicals (trends concentration/effect shown in Appendix A). The highest antiradical activity of the two phytochemicals was observed towards ABTS radical, with IC50 values in the order of tenths of mmol/L and without a statistical difference from Trolox confirmed also by TEACs equal to 1.083 and 0.812, respectively, for eugenol and capsaicin.

Eugenol was the only phytochemical able to inhibit the lipid peroxidation as α-tocopherol (no statistical difference between their IC50 values) with IC50s in the order of tens of mol/L. Vanillin was the molecule with the lowest antiradical and antilipoperoxidant activities (trends concentration/effect shown in Appendix A).

In the ORAC assay, all phytochemicals exerted an antiradical activity higher (* *p* < 0.05, ** *p* < 0.001) than Trolox (IC50 = 10.83 µM), with IC50 values in the order of units of µmol/L, demonstrating a strong activity in scavenging the AAPH radical after 55 min of exposure (Table 5). The fluorescent decay curves are shown in Appendix A.

For each phytochemical, Trolox equivalents (TEs) were calculated using slope values (m) obtained from calibration curves (Appendix A) drawn on whole reaction kinetics. The best ORAC value was shown by eugenol, followed by vanillin, although no statistical difference was observed among the three compounds.

## 4. Discussion

In this study, three methoxyphenol phytometabolites, eugenol, capsaicin, and vanillin, were tested to detect their antibacterial and antioxidant activities, which could be combined in applications in food preservation, following the hurdle concept.

Natural food preservatives may be antimicrobials and/or antioxidants, and could be used to help avoid foodborne diseases and/or oxidative cellular damage. Antimicrobial activity results highlighted that pathogens were more sensitive than spoilage bacteria to the compounds tested. All the phytochemicals were highly active against *S. aureus*, which is the most common cause of food poisoning among the pathogens considered. On the contrary, the phytochemicals were minimally active against the spoilage bacteria *L. plantarum*, which is one of the major spoilage bacteria, causing souring, off-flavor, discoloration, gas production, slime formation, and pH decrease [25]. These findings can be compared to those found in the literature. In fact, in 2010, Qiu and colleagues [26] observed that eugenol, at concentrations as low as 0.005 mM, was able to reduce the expression of virulence-related exoproteins in *S. aureus*. Similarly, Kalia et al. [27], Qiu et al. [28], and Oyedemi et al. [29] showed that capsaicin was active against the same strain at concentrations ranging from 0.08 to 0.83 mM, effect concentrations comparable to those of this study. Moreover, in 2009, Mourtzinos and collaborators [30] observed the antimicrobial properties of vanillin against *S. aureus* when it was exposed to a vanillin-vanillic acid mixture. On the other hand, the high resistance shown by *L. plantarum* to phytomolecules was also observed in 2010 by Bevilacqua and collaborators [31]. Basically, as considered by Ouwehand and collaborators [32], the sensitivity of Gram-positive potential pathogens combined with the relative resistance of lactobacilli to natural compounds could be an advantage for these latter in inhibiting the growth of potential pathogens while sparing the beneficial members of the intestinal microbiota.

Eugenol, capsaicin, and vanillin were similarly active towards *E. coli*, *S. putrefaciens*, and *S. aureus*; on the other hand, vanillin was statistically less active than the other two molecules against *P. aeruginosa*, as well as in comparison to eugenol against *L. plantarum* and *B. thermosphacta* (Appendix A).

Different mechanisms of action of the selected phytomolecules may be involved in the various antimicrobial activities observed. Eugenol can act directly on the cytoplasmic membrane, causing a non-selective permeability [33]; it can cause morphological-structural alterations, changing the composition in fatty acids [10,30]. It can also generate the production of intracellular ROS and inhibit the action of some bacterial enzymes, such as protease, histidine carboxylase, amylase, and ATPase [34,35]. Differently, capsaicin can induce the expression of an osmotic stress element as well as key genes for membrane biosynthesis and the PDR transporter network, and it can repress the genes for ribosomal components and for bacterial growth, as demonstrated by Kurita and co-authors [33]. Vanillin has a poorly understood mechanism of action [36,37].

In the literature, it has been reported that a strong antimicrobial activity of natural compounds may be attributed to the presence of free hydroxyl groups [38,39,40,41].

Eugenol, capsaicin, and vanillin have a hydroxyl group, nevertheless their activities probably differ due to the presence of functional groups at the end of their carbon side chain. Thus, the mere presence of the hydroxyl group is not always an indication of a strong antibacterial activity. Indeed, in the study by Pinheiro and collaborators [42], the presence of a functional group on the phenolic ring influenced the release of the H+ ion by the hydroxyl groups, conditioning the antimicrobial activity. In addition, Adeboye and collaborators [43] and Kemegne and co-authors [44] demonstrated that a different antimicrobial efficacy correlated with different lateral functional groups on the aromatic rings. The presence of unsaturated carbon–carbon bonds may influence the biological reactions and inhibitory activities of phenolic compounds in bacteria [38], explaining the broad-spectrum antibacterial activity shown by eugenol, which has a chemical structure with a carbon–carbon bond. In 2018, Hu and collaborators [45] found that eugenol is strongly active on several bacteria, including *Staphylococcus epidermidis*, *Staphylococcus carnosus*, *Salmonella enteritidis*, *Salmonella typhimurium*, *Streptococcus pneumoniae*, *Streptococcus mutans*, and *Listeria monocytogenes*, as well as on fungi belonging to the genera *Aspergillus* and *Penicillium*, and some yeasts. Catherine and co-authors [46] and Jeyakumar and Lawrence [47] also observed the antibacterial effect of eugenol on bacterial strains of food interest, such as *S. aureus*, *B. cereus*, and *E. coli*. To the best of our knowledge, only a few recent studies have reported the eugenol antimicrobial activity against food spoilage bacteria like *B. thermosphacta*, *S. putrefaciens*, *P. fluorescens*, *S. carnosus*, and *Lactobacillus* spp. [34,48,49,50]. In 2021, Takahashi and coauthors [50] stated that eugenol concentrations in the order of units of mmol/L were not able to control the growth of some *Lactobacilli*, including *L. plantarum*, confirming that *Lactobacilli* are less sensitive to eugenol, as found in the present study, where the MIC of *L. plantarum* was in the order of dozens of mmol/L.

The present study highlighted the strong antimicrobial efficacy of capsaicin on both foodborne pathogens and food spoilage bacteria, similar to that shown by eugenol (no statistical difference between the compounds) (Appendix A). Vanillin showed good antibacterial activity even though it does not have an unsaturated carbon–carbon bond. Probably, the antimicrobial activity could also be related to the guaiacol ring, which is present in all studied structures. Our results contribute to filling the gaps both in capsaicin antibacterial activity, for which few studies [12,27,28,29,51,52] have been carried out, and in vanillin, for which the activity on the selected spoilage bacteria has not been widely reported in literature [38,53]. In the present study, the spectrum of pathogen and spoilage bacteria tested has been amplified both in terms of species and in terms of assays performed, including data as MBC, LOAECs, and IC50s.

Apart from contamination by microbes, the oxidative damage is another crucial point in food preservation leading to health threats. Antioxidant activity results showed that eugenol was the most active molecule, followed by capsaicin and then vanillin. It is well known that methoxy and phenolic hydroxyl groups enhance the antioxidant power of phenolic compounds [54]. Hence, the guaiacol ring and the unsaturated carbon–carbon bond of the compounds tested in the present study may also play a role in the antioxidant activity as they do in the antimicrobial activity discussed above.

Eugenol was able to exert both strong antilipoperoxidant and antiradical activities (Appendix A), except for scavenging DPPH. Differently, capsaicin was able to exert better antiradical activity than antilipoperoxidant activity. In 2019, Lavorgna and co-authors [7] studied the free radical scavenging activity of capsaicin, obtaining IC50 values comparable to those obtained in this study (Table 4).

Comparative in vitro assays are essential to obtain relevant data on the antioxidant effects of phytochemicals because of their complex nature. According to Noguchi and Niki [55], it is essential to carry out tests based on different methods in order to better understand the behavior of the tested substances. Indeed, ABTS and DPPH assays are based on the generation of synthetic colored radicals or redox-active compounds without using physiological radicals with species that do not exist in the human body. In contrast, the ORAC assay detects chemical changes in a fluorescent molecule caused by a free radical attack reflecting physiological perturbations [56]. In the present study, the ORAC assay emphasized the radical scavenging power of the selected phytomolecules, also highlighting the activity of vanillin, with IC50 values at units of µmol/L, in the same order of magnitude as the other selected molecules (Table 5). Eugenol, capsaicin, and vanillin, as guaiacol derivatives (GDs), are able to exert strong free radical removal activity [57,58]. To date, two reaction antioxidant mechanisms have been suggested for guaiacolic derivatives: the transfer of hydrogen from the OH phenolic group and/or the transfer of the single electron from the DG to the radical in aqueous solutions [59] (here obtained in ABTS and ORAC assays); and the transfer of hydrogen from the phenolic group in non-polar solvents or in aqueous solutions pH ≤ 4 (TBARS assay). In detail, in 2020, Selles and colleagues [60] underlined that eugenol’s antiradical effectiveness may be explained by different mechanisms, such as the donation of hydrogen followed by the delocalization of the group substituted at the para-position, the dimerization between two phenoxylated radicals, and the complexation of diphenyl-picryl-hydrazyl radicals with an aryl radical. Furthermore, in 2012, Galano and Martínez [61] studied the mechanism of reaction used by capsaicin to scavenge free radicals, observing that the single electron transfer may be important for its reactions with •OH, •OCCl3, and •OOCCl3, confirming that the main process responsible for capsaicin peroxyl scavenging activity was found to be the hydrogen transfer from the OH phenolic group. Moreover, in 2011, Tai et al. [62] observed that an oxidative self-dimerization contributed to the total radical-scavenging ability of vanillin.

Since the antioxidant capacity is also defined as the ability of a compound to inhibit oxidative degradation, such as lipid peroxidation [63], the anti-lipoperoxidant activity was also evaluated in this study. Lipid peroxidation can be defined as the oxidative deterioration of lipids containing different carbon–carbon double bonds, often considered a toxicological phenomenon that can lead to various pathological aspects [64] because reactive aldehydes generated by lipid peroxidation can attack other cellular targets, such as proteins and DNA, and propagate the initial damage in cellular membranes to other macromolecules. In the present study, eugenol was the most antilipoperoxidant molecule, with a superimposable activity comparable to that of α-Tocopherol, and the observed activity was similar to that registered by Gulcin [65]. This study simplifies the understanding of the antioxidant power of the studied methoxyphenol phytometabolites via its statistical comparison to the standards. This allowed us to observe that eugenol and capsaicin have the same capacity as Trolox to scavenge the ABTS radical and that the three phytochemicals have a higher capacity to scavenge the APPH radical than that showed by Trolox.

Natural preservatives may exert antimicrobial and/or antioxidant activities, although these two properties are not always concurrently present. In the present study, three phytochemicals were selected and tested on the basis of their dual activity showing an antioxidant property starting from units of mg/L and an antimicrobial activity starting from dozens of mg/L. These results could be utilized in food industry applications in order to avoid the use of chemical additives. However, it would be necessary to consider the variety of factors present in complex food matrices that could diminish the efficacy of such compounds.

## 5. Conclusions

Access to sufficient amounts of safe and nutritious food is the key to sustaining life and promoting good health. The three guaiacol plant-derived compounds studied here showed both antimicrobial activities against foodborne pathogens and food spoilage bacteria, as well as antioxidant properties. All phytochemicals were highly active against the pathogen *S. aureus* but minimally active against the spoilage bacteria *L. plantarum*, and both eugenol and capsaicin were revealed to be broad-spectrum antibacterial molecules. Additionally, our results highlighted that eugenol exerted the highest antilipoperoxidant and antiradical activities, supporting its candidature as a potential natural food preservative. These outcomes will be useful for developing strategies to maintain food bioactivity with enhanced shelf life. Future applications could include the use of these nutraceuticals directly as food additives or applied as delivery systems in bioactive packaging with both antimicrobial and antioxidant properties.

## Figures and Tables

**Figure 1 foods-10-01807-f001:**
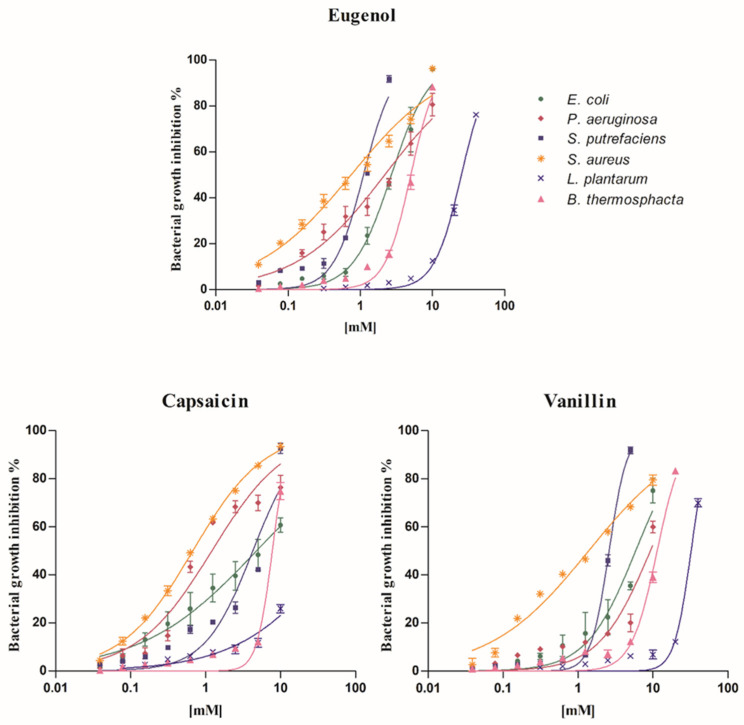
Concentration/response curves. Concentration/response curves of eugenol, capsaicin, and vanillin on *E. coli*, *P. aeruginosa*, *S. putrefaciens*, *S. aureus*, *L. plantarum*, and *B. thermosphacta* in the microbial susceptibility test. The trends are from the interpolation of three independent experiments, using GraphPad Prism 5 ((GraphPad Inc., San Diego, CA, USA). The bars represent the standard deviation.

**Figure 2 foods-10-01807-f002:**
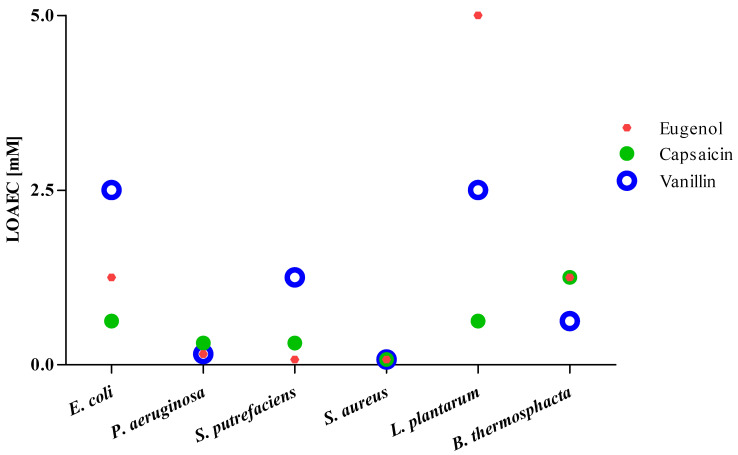
Lowest observed adverse effect concentration (LOAEC) values. LOAEC values [mM] after 24 h exposure (one-way ANOVA, Dunnett’s test of eugenol, capsaicin, and vanillin on the strains tested in the microbial susceptibility test).

**Table 1 foods-10-01807-t001:** Chemical information on eugenol, capsaicin, and vanillin.

Chemical Structure	Molecular Formula	Molar Mass (g/mol)	Chemical Name
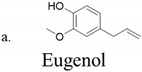	C_10_H_12_O_2_	164.2	4-Allyl-2-methoxyphenol
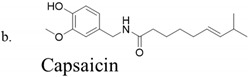	C_18_H_27_NO_3_	305.4	N-(4-Hydroxy-3-methoxybenzyl)-8- methylnon-trans-6-enamide
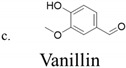	C_8_H_8_O_3_	152.1	4-Hydroxy 3-methoxybenzaldehyde

**Table 2 foods-10-01807-t002:** Median concentrations of bacterial growth inhibition (IC50s). IC50 values, coming from the interpolation of three independent experiments, are expressed in mM with a 95% confidence range (in brackets). Letters indicate the significant differences among different strains for each phytochemical, one-way ANOVA, Tukey’s multiple comparison test (*p* < 0.05).

Bacterial Strain	IC50		
	Eugenol	Capsaicin	Vanillin
*E. coli*	2.70 ^a,b^ (2.39–3.06)	4.79 ^a,b^ (3.46–6.63)	5.87 ^b,c^ (4.72–7.31)
*P. aeruginosa*	2.19 ^a,b^ (1.76–2.72)	1.21 ^a^ (0.92–1.60)	9.23 ^c,d^ (6.99–12.20)
*S. putrefaciens*	1.11 ^a^ (0.97–1.28)	4.28 ^a,b^ (3.29–5.55)	2.60 ^a,b^ (2.50–2.71)
*S. aureus*	0.75 ^a^ (0.60–0.93)	0.68 ^a^ (0.64–0.72)	1.38 ^a^ (1.17–1.63)
*L. plantarum*	25.20 ^c^ (24.01–26.45)	>10	32.31 ^e^ (30.20–34.57)
*B. thermosphacta*	5.02 ^b^ (4.65–5.42)	7.77 ^b^ (7.18–8.39)	11.38 ^d^ (10.46–12.37)

**Table 3 foods-10-01807-t003:** Minimal inhibitory concentration (MIC) and minimal bactericidal concentration (MBC) in mM of eugenol, capsaicin, and vanillin on the strains tested.

Bacterial Strain	Eugenol		Capsaicin		Vanillin	
	MIC	MBC	MIC	MBC	MIC	MBC
*E. coli*	12.5	25	>12.5	>12.5	12.5	25
*P. aeruginosa*	12.5	25	>12.5	>12.5	25	50
*S. putrefaciens*	3.125	6.25	12.5	>12.5	6.25	12.5
*S. aureus*	6.25	12.5	12.5	>12.5	12.5	25
*L. plantarum*	50	100	>12.5	>12.5	100	>100
*B. thermosphacta*	12.5	25	>12.5	>12.5	25	50

**Table 4 foods-10-01807-t004:** IC50 and TEAC values. Median effective concentrations (IC50s, in mM) of phytochemicals able to scavenge the DPPH or ABTS radicals, or to inhibit the lipid peroxidation. * *p* < 0.05, *** *p* < 0.0001). Letters indicate the significant differences among phytochemicals for each assay (one-way ANOVA, Tukey’s multiple comparison test). All data come from three independent experiments. The 95% confidence limits are shown in parentheses.

	IC50			TEAC	
	DPPH	ABTS	TBARS	DPPH	ABTS
Trolox	0.016 (0.014–0.019)	0.013 (0.011–0.015)	-	-	-
α-Tocopherol	-	-	0.013 (0.011–0.014)	-	-
Eugenol	0.152 ^a,^*** (0.140–0.165)	0.012 ^a^ (0.010–0.014)	0.024 ^a^ (0.019–0.030)	0.105	1.083
Capsaicin	0.090 ^a,^* (0.088–0.092)	0.016 ^a^ (0.014–0.017)	0.198 ^b,^*** (0.154–0.251)	0.178	0.812
Vanillin	3.199 ^b,^*** (2.638–3.879)	5.566 ^b,^*** (4.663–6.642)	0.802^c,^*** (0.752–0.855)	0.005	0.002

**Table 5 foods-10-01807-t005:** Oxygen radical absorbing capacity (ORAC) assay results. IC50 values (µM) with a 95% confidence range (in brackets) come from the interpolation of three independent experiments. ORAC values, expressed as µM Trolox with equivalent activity to 1 µM phytochemical (or Trolox equivalents: µM TE), are reported as the mean of three independent experiments with standard deviation. Asterisks highlight significant differences from Trolox (one-way ANOVA, Dunnett’s test: * *p* < 0.05, ** *p* < 0.001). Letters indicate significant differences among phytochemicals for each outcome for *p* < 0.05 (one-way ANOVA, Tukey’s multiple comparison test).

	IC50	ORAC Value (TE)
Trolox	10.83 (10.33–11.35)	-
Eugenol	4.11 ^a,^** (3.91–4.32)	2.12 ± 0.08 ^a^
Capsaicin	5.81 ^a,^** (5.40–6.24)	1.60 ± 0.25 ^a^
Vanillin	6.15 ^a,^* (5.53–6.83)	1.81 ± 0.19 ^a^

## Data Availability

Not applicable.

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
