# Peer review of "Natural Methoxyphenol Compounds: Antimicrobial Activity against Foodborne Pathogens and Food Spoilage Bacteria, and Role in Antioxidant Processes"

_foods, 2021, doi:10.3390/foods10081807_

Round 1

Reviewer 1 Report

This manuscript tries to evaluate the antibacterial and antioxidant activities of three methoxyphenol phytometabolites, eugenol, capsaicin and vanillin. In fact, the antibacterial activities of eugenol, capsaicin and vanillin have been widely studied in the literature also on both food-borne as well as food spoilage bacterial strains. Some previous reports which addressed this field have been mentioned in the introduction and discussion sections in this manuscript as well. Otherwise, the antioxidant activities of eugenol, capsaicin and vanillin have been documented as well. Most of all, most of the important findings in the present manuscript can be found or interpreted from previous published reports. In this manuscript, the presentations and interpretations of data are sometimes confusion, especially in the explanations about superscripts within tables. The discussions are not discussed in depth or in a meaningful way to could compare the current findings with previous known ones to could emphasize and focus the new information here. As described in the abstract section, the authors indicated that “This study will be useful in future applications of these guaiacol derivatives in food preservation against microbial and oxidative deterioration.” This information can be obtained through the current data by the literature. I encourage the authors compare and cite more previous reports to could strengthen their findings. In general, the results obtained are methodically validated in part, and I have some critical comments for improving the MS. 

Line 44 Table 1: the descriptions for the title of table 1 seems redundant. Please simplify.

Line 53: introduction. The authors indicate that “To the best of our knowledge, although these selected molecules have been known to possess antimicrobial properties [9-12], to date, the differences in their activities against food-borne pathogens and food spoilage bacteria have not been investigated yet.” In fact, the antibacterial activities of eugenol, capsaicin and vanillin against both food-borne and food-spoilage bacterial strains have been widely investigated in the literature. I recommend the authors cited more previous reports and emphasize your new and important findings in the introduction in a clear way.

Line 144-146: the descriptions about using resazurin to determine the MIC values are not clear. Please add more explanations about how to conduct this experiment.

Line 242 table2. The definition about superscripts here are not clear enough. Please clarify or revise.  

Line 268  it may be better to describe the “Gram- or Gram+” as “Gram–negative and Gram-positive”.

Line 330  table 3 shows the MIC and MBC results about three phytochemicals against pathogens. But you indicates that “Vanillin did not show neither bacteriostatic nor bactericidal high activity.” However, according to the findings in table 3 or figure 1, vanillin indeed displays dose-dependent antibacterial activities against the selected pathogens. Please clarify.

Line 342 table 4;  the descriptions about the superscripts are confusion. It is not easy to follow the exact meanings about those statistically results and meanings here. Please clarify or revise.

Line 332-356 the explanations about the main findings from the experiments are not clear. The authors should cite the exact information in the suitable texts to let the readers understand these findings. The figure S12, S2, S2 are not cited in the proper positions. This section is not easy to follow.

Line 362-370 table 5: the legends for this table is not clear. Too much unclear information is described here. I recommend the authors explains more about their findings and interpretations regarding the contents in table 5.

Line 404 Table 6; it seems to me this table 6 is redundant because similar findings can be obtained from figure 1 and table 1.

Line 386 discussion: The section is not discussed in depth and the authors need to compare the current findings with previous reported ones to could emphasize the new information here. To show the findings from previous studies seems not be enough since several previous reports have conducted similar experiments as shown in this manuscript. Please add more explanations and comparisons to previous reports to could provide new information for the readers.

Author Response

We thank the reviewers for the great job they did. After their suggestions the MS has definitely improved.

Reviewer 1:

This manuscript tries to evaluate the antibacterial and antioxidant activities of three methoxyphenol phytometabolites, eugenol, capsaicin and vanillin. In fact, the antibacterial activities of eugenol, capsaicin and vanillin have been widely studied in the literature also on both food-borne as well as food spoilage bacterial strains. Some previous reports which addressed this field have been mentioned in the introduction and discussion sections in this manuscript as well. Otherwise, the antioxidant activities of eugenol, capsaicin and vanillin have been documented as well. Most of all, most of the important findings in the present manuscript can be found or interpreted from previous published reports. In this manuscript, the presentations and interpretations of data are sometimes confusion, especially in the explanations about superscripts within tables. The discussions are not discussed in depth or in a meaningful way to could compare the current findings with previous known ones to could emphasize and focus the new information here. As described in the abstract section, the authors indicated that “This study will be useful in future applications of these guaiacol derivatives in food preservation against microbial and oxidative deterioration.” This information can be obtained through the current data by the literature. I encourage the authors compare and cite more previous reports to could strengthen their findings. In general, the results obtained are methodically validated in part, and I have some critical comments for improving the MS.

R.: In the abstract the sentence “This study will be useful in future applications of these guaiacol derivatives in food preservation against microbial and oxidative deterioration.” has been changed into: This study, comparing the antimicrobial and antioxidant activities of three guaiacol derivatives, enhances their use in future applications as food additives for contrasting both common pathogens and spoilage bacteria and for improving the shelf life of preserved food.

Line 44 Table 1: the descriptions for the title of table 1 seems redundant. Please simplify.

R.: The title of Table 1 has been changed in: Chemical information on eugenol, capsaicin and vanillin

Line 53: introduction. The authors indicate that “To the best of our knowledge, although these selected molecules have been known to possess antimicrobial properties [9-12], to date, the differences in their activities against food-borne pathogens and food spoilage bacteria have not been investigated yet.” In fact, the antibacterial activities of eugenol, capsaicin and vanillin against both food-borne and food-spoilage bacterial strains have been widely investigated in the literature. I recommend the authors cited more previous reports and emphasize your new and important findings in the introduction in a clear way.

R.: The introduction has been improved and the aim has been clarified [new lines 50-53 and 62-72].

Line 144-146: the descriptions about using resazurin to determine the MIC values are not clear. Please add more explanations about how to conduct this experiment.

R.: the description on how MIC and MBC were calculated has been improved.

Line 242 table2. The definition about superscripts here are not clear enough. Please clarify or revise. 

R.: The caption and the superscripts have been simplified and numbers have been deleted to improve the reader comprehension.

Table 2: Median concentrations of bacterial growth inhibition (IC50s). IC50 values, coming from the interpolation of three independent experiments, are expressed in mM with a 95% confidence range (in brackets). Letters indicate the significant differences among different strains for each phytochemical, One Way-ANOVA, Tukey's Multiple Comparison Test (p<0.05).

Line 268  it may be better to describe the “Gram- or Gram+” as “Gram–negative and Gram-positive”.

R.: Done

Line 330  table 3 shows the MIC and MBC results about three phytochemicals against pathogens. But you indicates that “Vanillin did not show neither bacteriostatic nor bactericidal high activity.” However, according to the findings in table 3 or figure 1, vanillin indeed displays dose-dependent antibacterial activities against the selected pathogens. Please clarify.

R.: Thank you for this comment. It was a mere error because the sentence was referred to L. plantarum.

Line 342 table 4;  the descriptions about the superscripts are confusion. It is not easy to follow the exact meanings about those statistically results and meanings here. Please clarify or revise.

R.: The caption and the superscripts have been simplified and numbers have been deleted. The caption changed into:

Table 4. IC50 and TEAC values. Median effective concentrations (IC50s, in mM) of phytochemicals able to scavenge DPPH and ABTS radicals and to inhibit the lipid peroxidation (TBARS). For DPPH and ABTS the results are also expressed as Trolox Equivalent Antioxidant Capacity (TEAC). Significant differences from standards (Trolox or α-Tocopherol) are highlighted by asterisks (One Way-ANOVA, Dunnett’s test: *p<0.05, *** p < 0.0001). Letters indicate the significant differences among phytochemical for each assay (One Way-ANOVA, Tukey's Multiple Comparison Test). All data come from three in-dependent experiments. In brackets 95% confidence limits.

Line 332-356 the explanations about the main findings from the experiments are not clear. The authors should cite the exact information in the suitable texts to let the readers understand these findings. The figure S12, S2, S2 are not cited in the proper positions. This section is not easy to follow.

R.: after the modification of Table 4 caption and the improvements going from the new line 586-649, the text is more clear.

Supplementary figures have been moved in a more suitable position.

Line 362-370 table 5: the legends for this table is not clear. Too much unclear information is described here. I recommend the authors explains more about their findings and interpretations regarding the contents in table 5.

R.: The caption and the whole table 5 were deeply simplified. Detailed information were moved to supplementary material.

Line 404 Table 6; it seems to me this table 6 is redundant because similar findings can be obtained from figure 1 and table 1.

R.: Table 6 and 7 have been moved into the supplementary material as Tables S1 and S2.

Line 386 discussion: The section is not discussed in depth and the authors need to compare the current findings with previous reported ones to could emphasize the new information here. To show the findings from previous studies seems not be enough since several previous reports have conducted similar experiments as shown in this manuscript. Please add more explanations and comparisons to previous reports to could provide new information for the readers.

R.: This section has been improved providing the new information coming from the present study.  

Reviewer 2 Report

Basically, the approach taken in the study is interesting, and data are appropriate. The obtained results are very interesting and promising, but I have some questions, comments and remarks: 

  1. On what basis were the quantitative ranges of the active substances tested?
  2. In Table 1, it is worth adding the molar mass of the compounds tested.
  3. There is no unit given in tables 2, 3 and 4.
  4. Instead of Gram - and Gram +, it should be Gram (-) and Gram (+).
  5. The discussion should also include information about the cytotoxicity of the active compounds tested.
  6. There should be no literature citation in the conclusions section.

Author Response

We thank the reviewers for the great job they did. After their suggestions the MS has definitely improved.

Reviewer  2:

Basically, the approach taken in the study is interesting, and data are appropriate. The obtained results are very interesting and promising, but I have some questions, comments and remarks:

On what basis were the quantitative ranges of the active substances tested?

R.: The ranges were chosen on the basis of phytochemicals solubility and of the range finding           tests as reported in the text

In Table 1, it is worth adding the molar mass of the compounds tested.

R.: the molar mass has been added to Table 1

There is no unit given in tables 2, 3 and 4.

R.: Units were added.

Instead of Gram - and Gram +, it should be Gram (-) and Gram (+).

R.: Gram -  and Gram + have been changed into Gram-negative and Gram-positive.

The discussion should also include information about the cytotoxicity of the active compounds     tested.

R.: The cytotoxicity is a topic of great interest for us but it is not in the experimental design            of the present study

There should be no literature citation in the conclusions section.

R.: Literature citations were removed by the conclusions.

Round 2

Reviewer 1 Report

The revised version of the manuscript has been considerately improved.